# SVR-Net: A Sparse Voxelized Recurrent Network for Robust Monocular SLAM with Direct TSDF Mapping

**DOI:** 10.3390/s23083942

**Published:** 2023-04-13

**Authors:** Rongling Lang, Ya Fan, Qing Chang

**Affiliations:** School of Electronics and Information Engineering, Beihang University, Beijing 100191, China

**Keywords:** monocular SLAM, deep learning, TSDF mapping

## Abstract

Simultaneous localization and mapping (SLAM) plays a fundamental role in downstream tasks including navigation and planning. However, monocular visual SLAM faces challenges in robust pose estimation and map construction. This study proposes a monocular SLAM system based on a sparse voxelized recurrent network, SVR-Net. It extracts voxel features from a pair of frames for correlation and recursively matches them to estimate pose and dense map. The sparse voxelized structure is designed to reduce memory occupation of voxel features. Meanwhile, gated recurrent units are incorporated to iteratively search for optimal matches on correlation maps, thereby enhancing the robustness of the system. Additionally, Gauss–Newton updates are embedded in iterations to impose geometrical constraints, which ensure accurate pose estimation. After end-to-end training on ScanNet, SVR-Net is evaluated on TUM-RGBD and successfully estimates poses on all nine scenes, while traditional ORB-SLAM fails on most of them. Furthermore, absolute trajectory error (ATE) results demonstrate that the tracking accuracy is comparable to that of DeepV2D. Unlike most previous monocular SLAM systems, SVR-Net directly estimates dense TSDF maps suitable for downstream tasks with high efficiency of data exploitation. This study contributes to the development of robust monocular visual SLAM systems and direct TSDF mapping.

## 1. Introduction

A simultaneous localization and mapping (SLAM) system localizes the agent in the environment and reconstructs the environment. It is fundamental to other tasks, such as navigation and planning, etc. In visual SLAM, the agent senses the environment using a monocular, stereo or RGBD camera, and monocular SLAM is the most challenging due to difficulties in robust pose estimation and map construction. Moreover, the demand for dense maps in many applications presents a new set of challenges that must be addressed by existing monocular SLAM methods.

Typically, the pose of the camera is represented by a quantity in the special Euclidean group SE(3), which includes components of translation and rotation. The map in SLAM is often represented by a depth map, which contains the distance between each environmental point corresponding to each pixel in the image and the camera. Traditional SLAM methods are classed as either direct or indirect, according to the way they use information. Direct methods solve for poses and depths through photometric error. Photometric error utilizes abundant information such as lines and intensity variations, but complex patterns of images can cause local minima when optimizing photometric error. The optimization process is also unstable when an image is noisy. Indirect approaches exploit information from feature points. Feature points locate sparse local regions of images and compute descriptors which are similar from different perspectives. Reprojection error is optimized to minimize the distance of projection and matched location of feature points. For example, ORB-SLAM [1] uses a sparse ORB feature for tracking and mapping, which can achieve high performance when feature points are correctly matched. However, in regions with few or repetitive textures, feature points are often erroneously detected or matched, which also leads to a robustness problem.

With the rapid growth of deep learning, neural networks have been successfully applied in image processing due to their powerful feature extraction capability [2,3]. They have also been adopted in SLAM to tackle the robustness problem. Some work has studied replacing submodules of SLAM with networks. In high-level feature extraction, using networks can provide an advantage in computing feature points [4,5,6,7,8] or finding good correspondences [9,10]. Learning-based outlier rejection is explored to produce robust matches [11,12,13]. Other work has investigated performing end-to-end learned SLAM [14,15,16,17]. After training on a wide variety of scenes with ground-truth poses, the networks can utilize prior information to avoid catastrophic errors.

However, previous end-to-end SLAM networks are not trained on applicable dense maps, which limits their availability. Networks such as TartanVo [16] and DROID-SLAM [15] only provide pose results. Other networks including DeepV2D [14] use depth maps or point clouds as map representation, which is not enough for downstream tasks. Post-processing and fusion could directly use depth maps or point clouds to produce maps suitable for downstream tasks, but additional information extracted from the networks would be omitted.

Instead of depth maps or point clouds, denser representation of environment geometry is considered, which is typically required to enable downstream tasks. To reconstruct a dense scene from a set of calibrated views, traditional multi-view stereo techniques rely on local depth map computation and global depth map fusion. The Truncated Signed Distance Function (TSDF) is a fusion format that contains distance information useable for navigation and planning. TSDF assigns each 3D point a signed distance from the nearest environment surface. The sign is decided by whether the point is hidden by the surface, and distances with absolute values greater than 1 are truncated. The result of two-stage reconstruction is prone to being either layered or scattered due to depth inconsistency [18]. Thus, deep neural architectures regressing TSDF for direct 3D reconstruction were designed in recent studies [18,19,20]. However, these methods require the camera poses as prior information, and they cannot guarantee the quality of maps when pose estimation is not accurate. In monocular SLAM, map construction is more challenging because there are more uncertainties in pose estimation.

In this work, we propose a monocular SLAM system with direct TSDF mapping based on our sparse voxelized recurrent network, SVR-Net. For a sequence of monocular image frames, SVR-Net outputs both poses and a TSDF map. The network takes an end-to-end pipeline that extracts features for correlation and recursively matches them, which is shown to have STOA localization performance in DROID-SLAM. Moreover, to enable TSDF reconstruction in the SLAM pipeline, a novel matching network is designed using sparse 3D convolution on voxels, combining the recursive matching with the TSDF mapping branch. This study makes a contribution towards the advancement of robust monocular visual SLAM systems and direct TSDF mapping.

The main features of the proposed method are summarized as follows:(1)Robust monocular SLAM. SVR-Net’s semantic encoder encodes information such as the scale of scenes to guide matching, which helps monocular pose estimation. SVR-Net utilizes correlation operations to reduce both the size of features and dependence on a specific scene’s semantic information, which avoids overfitting. After end-to-end training on ScanNet [21] with ground truth poses and maps, SVR-Net successfully estimates poses for all nine scenes of the challenging TUM-RGBD [22] benchmark, whereas ORB-SLAM fails for most of them.(2)Accurate pose estimation. Iterative updates are carried out in a recurrent network to search for the optimal match. Gauss–Newton updates are embedded in iterations to impose geometrical constraints, which improves the accuracy of pose estimation. Experimental results using the TUM-RGBD benchmark show that the pose accuracy of SVR-Net is comparable to that of DeepV2D.(3)Direct TSDF mapping. SVR-Net directly regresses the TSDF values and occupancy confidences of given voxels. Unlike previous monocular SLAM systems that produce depth maps or sparse 3D points, SVR-Net produces dense TSDF maps suitable for downstream tasks including navigation and planning. Compared with TSDF reconstruction methods that take depth maps as intermediate representation, direct TSDF mapping avoids depth inconsistency. Moreover, SVR-Net’s direct TSDF mapping is more data-efficient because both pose and map are estimated using the same features.

## 2. Related Works

Monocular SLAM uses monocular visual information to locate the carrier and model the environment. Traditional methods are classified as either direct or indirect methods according to the way information is used [23]. Indirect approaches first detect and match key points between two frames, then predict poses and 3D points by minimizing the reprojection error [1,24]. Direct approaches operate directly on pixel intensities, and they estimate poses and depths through photometric error [23,25,26]. However, both direct and indirect methods are vulnerable to outliers.

Deep learning has been proposed to improve the robustness of SLAM systems. Some work has investigated replacing hand-crafted with learned features [4,5,6,7,8], using neural 3D representations [27,28,29,30,31,32,33], and combining learned energy terms with classical optimization backends [34,35]. In other work, researchers have tried to produce end-to-end learning for SLAM or VO systems [14,15,16,17]. Among these methods, DROID-SLAM consists of end-to-end recurrent iterative updates of camera pose and pixel-wise depth. It achieves large improvements in accuracy and substantially reduces catastrophic failures. Sparse voxel-based networks have been applied for the efficient processing of point clouds and voxels [36,37,38]. SVR-Net contains an iterative pipeline that is similar to DROID-SLAM, but it estimates TSDFs, instead of depth values, with a sparse voxelized structure for direct mapping.

The implicit representations of surface geometry through SDF are introduced in [39]. The efficacy of TSDFs in fusing high-rate, noisy depth data from consumer-grade depth cameras, which was first demonstrated by Newcombe et al. [40], has led to a rapid rise in their popularity. Furthermore, this representation has led to their recent adoption on robotic platforms [41,42,43], where the distance field has shown additional utility for optimization-based motion planning [44,45].

Recent studies have shown the advantage of deep networks in TSDF mapping, compared with traditional multi-view methods. NeuralRecon directly reconstructs local surfaces, represented as sparse TSDF volumes for each video fragment sequentially, by a neural network [18]. VoRTX is occlusion-aware, leveraging the transformer architecture to predict an initial, projective scene geometry estimate [20]. VolumeFusion replicates the traditional two-stage framework with deep neural networks, improving both the interpretability and the accuracy of the results [46]. The suggested network is closely related to TSDF mapping networks, but does not need camera poses to be provided. Table 1 shows a comparison of existing learning-based localization and mapping methods.

## 3. Method

The SLAM system employs a coarse-to-fine strategy to achieve tracking and global dense mapping, as shown in Figure 1. At the first stage, the raw pose and local map of a pair of input frames are estimated using SVR-Net. The map is represented by sparse voxels with TSDF values. Then, the map is fused with the first-stage global map. At the second stage, the voxels are up-sampled, halving their intervals. Subsequently, the same network is utilized to refine the pose and map. The resulting fine local map is fused with the second-stage global map, and SVR-Net proceeds to track the next pair of frames.

SVR-Net is an end-to-end network for monocular simultaneous tracking and mapping. Its input consists of a pair of RGB frames with a query set of voxel coordinates. It outputs the relative pose $T_{n}$ of the frames and TSDF values $d_{n}$ of the voxels. For each pair of frames, the earlier frame is designated as the keyframe, and the later frame is designated as the reference frame. As shown in Figure 2, SVR-Net first extracts the image features into 2D feature maps. The feature map of the key frame is then transformed into feature voxels and correlated with the features of the reference frame. After sampling based on the current pose estimation, a matching network iteratively matches the features and updates the estimation of pose and map. The iterative pipeline performs a search in a correlation field of the features to find the optimal match, thereby enhancing both accuracy and robustness.

### 3.1. Voxel Feature Extraction and Correlation

In the feature extraction, deep features of the key frame are computed and back-projected into 3D voxels. Then, correlation maps between features of the voxels and the reference frame are computed to provide similarity information for matching.

#### 3.1.1. Voxel Feature Extraction

2D feature maps are extracted from the input images using 2D convolution networks, namely, a metric encoder and a semantic encoder. Both networks produce feature maps at 1/8 the input image resolution. The metric encoder is a subnet of MnasNet [47] and contains two inverted residual blocks. The metric features of both frames are used to build correlation between voxels and images. The semantic encoder resembles the context network in RAFT, which consists of six residual blocks. The reference frame is only fed into the metric encoder to produce a metric map $I_{g}^{{}^{\prime }}$ for feature correlation, while the semantic features of the key frame provide the initial hidden state and inputs for recurrent feature matching.

Given voxel coordinates, both the semantic and metric feature maps of the key frame are back-projected to produce the 3D feature volume. Assume that there are *N* input voxels with their coordinates $X_{i},1\leq i\leq N$. For each voxel coordinate, let the projected coordinate be $x_{i}$.
(1)$$x_{i}=\Pi \left(T_{k}X_{i}\right)$$

Here, Π is the projection function of the camera that maps 3D points onto the image. $T_{k}$ is the transformation matrix of the key frame. Bilinear interpolation is carried out at $x_{i}$ on the feature maps of the key frame to produce voxelized features of $X_{i}$. To exploit sparsity, the coordinates and features are stored in separate matrices X, $F_{s}$ and $F_{g}$ with N rows. Here, the subscript *s* and *g* represent semantic features and metric features, respectively.

#### 3.1.2. Voxel Feature Correlation

After the voxelization of features from the key frame, the feature correlation between the key frame and the reference frame becomes feature correlation between the feature voxels and the reference frame’s metric feature map. The correlation maps are computed with dot products within the metric features. For the *i*th voxel, its correlation map at the first layer is $C_{i}^{1}$.
(2)$$C_{i}^{1}\in R^{h\times w},C_{ikl}^{1}=\sum_{j=1}^{D}F_{g,ij}I_{g,klj}^{{}^{\prime }}$$

Here, h,w, and *d* are the height, width, and feature dimensions of the metric feature map. $F_{g,ij}$ is the *j*th value of the voxel’s metric feature. $I_{g,klj}^{{}^{\prime }}$ is the *j*th value of the reference frame’s metric feature at position (k,l). The feature correlation reduces feature dimensions and provides scene-independent information to the downstream networks, which can improve generalization.

To present similarity information at different scales, the $C_{i}^{1}$ is pooled with kernel size 2,4,8. All generated maps are $C,C=\left\{C_{i}^{m},1\leq i\leq N,1\leq m\leq 4\right\}$. $C_{i}^{m}$ has a resolution of $\frac{h}{2^{m-1}}\times \frac{w}{2^{m-1}},m=1,2,3,4$. The four layers of the correlation maps form a pyramid where the correlation field is sampled at each iteration.

### 3.2. Sparse Recurrent Feature Matching

Feature matching uses an iterative pipeline to estimate the correspondence between the voxels and the reference frame. The voxels are first projected to the reference frame using the estimated pose from the last iteration. Then, correlation values sampled from projection coordinates are fed into the matching network, which predicts both updated TSDF values and corrected projection coordinates.

#### 3.2.1. Sampling

In the sampling operation, correlation vectors are generated for all voxels and their projection coordinates on correlation maps, as demonstrated in Figure 3. For each correlation map of each projection coordinates $x_{i}$, a grid with radius *r* is applied. Every point of the grid corresponds to an offset from the coordinate where the correlation value is sampled with bilinear interpolation. For all four correlation maps of the voxel, the values from four grids are concatenated to generate the correlation vector. All correlation vectors are stacked to a matrix $C_{T}\in R^{N\times 4{\left(2r+1\right)}^{2}}$. The subscript T, which represents the pose of the reference frame, varies during iteration and results in different correlation values.

#### 3.2.2. Matching Network

The coordinates and features of voxels stored in *N*-row matrices are fed into the matching network. The network produces three outputs: (1) revisions of voxels’ TSDF values $\Delta d\in R^{N}$; (2) corrections of projection coordinates $\Delta x\in R^{N\times 2}$; and (3) confidence weights $w\in R_{+}^{N\times 3}$. Each row of the outputs corresponds to each input voxel. The first two columns of confidence weights represent the confidence of the projection coordinates. The last column is the confidence of voxel occupancy, whose value is between 0 and 1.

The matching network is implemented with sparse 3D convolution to process voxels with varying spatial distributions and sparsity. A gated recurrent unit (GRU) is also adopted to support iteration. The formulation of a GRU layer is as follows.
(3)$$\begin{array}{c} Z_{t}=\sigma \left(SConv_{3}\left(\left[H_{t-1},G_{t}\right]\right)\right)\\ R_{t}=\sigma \left(SConv_{3}\left(\left[H_{t-1},G_{t}\right]\right)\right)\\ {\tilde{H}}_{t}=\tanh\left(SConv_{3}\left(R_{t}⨀G_{t}\right)\right)\\ H_{t}=\left(1-Z_{t}\right)⨀H_{t-1}+Z_{t}⨀{\tilde{H}}_{t} \end{array}$$

Here, ⨀ is element-wise multiplication and σ is sigmoid activation. Sparse 3D convolution, $SConv_{3}$, implicitly utilizes voxel coordinates to fuse features. The hidden state from the last iteration is denoted by $H_{t-1}$. The semantic feature $F_{s}$ is split into $F_{s,h}$ and $F_{s,i}$, to provide the initial hidden state and input feature, respectively. The input feature $G_{t}$ of the GRU is a concatenation of these features: (1) the semantic feature for input $F_{s,i}$; (2) a correlation feature $F_{c}$; and (3) a motion feature $F_{m}$. The correlation feature is obtained by encoding $C_{T}$ with two $SConv_{3}$ layers. The motion feature is obtained by encoding a motion pattern matrix M with a similar network, which is informative for GRU to perceive its progress. The motion pattern consists of flows and errors of projection coordinates, as demonstrated in Section 3.3.

The remaining components of the matching network are three branches to produce outputs at iteration *t*: $\Delta d_{t}$, $\Delta x_{t}$ and $w_{t}$. Each branch contains two $SConv_{3}$ layers and takes $H_{t}$ as common input, which effectively utilizes information in features.

### 3.3. Iterative Tracking and Mapping

The outputs of feature matching are utilized to update the estimations of the pose and map in each iteration of SVR-Net. A local map is obtained from the final iteration. Then, map fusion of the SLAM pipeline integrates the local maps from all input frames to enhance global consistency.

#### 3.3.1. Pose Estimation with Gauss–Newton Update

After each iteration of feature matching, the estimated projection coordinates from the last iteration are corrected with Δx. Then, under the constraints of the projection coordinates and the confidence weights, the classical re-projection error *E* is optimized using Gauss–Newton iteration to produce an updated pose estimation.
(4)$$E=\sum_{i=1}^{N}e_{i}^{T}W_{i}e_{i}$$

Here, $e_{i}$ is the re-projection error from the *i*th voxel. $W_{i}$ is the corresponding weight matrix based on the confidence weights w.
(5)$$\begin{array}{c} e_{i}=x_{i}+\Delta x_{i}-\Pi \left(TX_{i}\right)\\ W_{i}=diag\left(w_{i1},w_{i2}\right) \end{array}$$

The Gauss–Newton update on the pose T is performed within Lie algebra. The update is differential with respect to Δx, making it possible to train the entire network end-to-end.

The motion pattern of the updated pose is informative to the matching network because the network can learn to adjust its matching strategy reactively at each iteration. The motion pattern M contains the flows and errors of all voxels. Let the initial pose before the first iteration be $T_{0}$ and the estimated pose at iteration *t* be $T_{t}$. Then, the flow of the *i*th voxel is defined as $\Pi \left(T_{t}X_{i}\right)-\Pi \left(T_{0}X_{i}\right)$, which informs the corresponding optical flow on the image. The error is the re-projection error of $T_{t}$.

#### 3.3.2. Map Update and Fusion

For each pair of frames, a local map within the key frame’s view frustum is estimated. The map is represented as a set of voxels with TSDF values and occupancy confidences. The coordinates of the voxels are initialized, filling the view frustum with a maximum depth. The distance between neighboring coordinates, i.e., voxel size, controls the resolution of the map. However, when the voxel size is small, the huge number of voxels can cause a heavy memory burden on the GPU. This problem is addressed using the coarse-to-fine strategy.

At the first stage, the TSDF values $d_{0}$ in the view frustum are initialized with zero. With iterative running of the matching network, the TSDF values are successively added with revisions Δd. After the final iteration of the network, the first-stage TSDF estimations are obtained with the confidence weights w. Occupancy confidences $w_{o}$ are the last column of the weights, which are used in map fusion. After the first-stage map fusion, the voxels in the view frustum are up-sampled and fed into SVR-Net again for second-stage estimation, producing a fine-grained map.

As the frames are input in sequence in the SLAM pipeline, all local maps are fused together to produce a global map. The global map is computed using a linear weighting method similar to TSDF integration [40]. The global map is initialized with the local map of the first key frame. For each subsequent local map, the global TSDF values of overlapped voxels are averaged according to global and local occupancy confidences. Then, the global occupancy confidences are updated in a cumulative manner. The data of voxels in a new region are directly attached to the global map.

## 4. Experiments

SVR-Net is trained on ScanNet(V2), and the full SLAM system is evaluated using TUM-RGBD. The ScanNet dataset includes 1613 indoor scenes with ground-truth camera poses and depth maps. The TUM-RGBD dataset contains nine indoor scenes with ground-truth poses. Without depth maps, the RGB images in TUM-RGBD contain heavy motion blur, which is a challenge for monocular SLAM.

ScanNet is split into a training set and a validation set, with the training set containing 1513 scenes. In order to improve tracking accuracy with various motion patterns, each pair of training frames is sampled using a random selection strategy from a successive subsequence. The length of each subsequence is controlled to satisfy the condition that there are exactly nine frames with mutual distance greater than thresholds. The distance threshold of rotation is 15°, and the threshold of translation is 0.1 m. The ground-truth TSDF global maps are generated by integrating depth frames of the training sequences. For each sequence, there are two scales of ground-truth maps, corresponding to two stages. The voxel sizes of the two scales are 16 cm and 8 cm. For training, the ground-truth map of each image frame is set to the part of the global map within the frame’s view frustum. Training supervision includes both pose loss and map loss. The pose loss measures the distance between ground truth and the predicted pose, $L_{pose}={∥Log_{SE3}\left(T^{-1}T_{gt}\right)∥}_{2}$. The map loss is the mean difference between the TSDF values, $L_{map}={∥d-d_{gt}∥}_{1}$.

The hardware used for training and evaluation includes Intel Xeon Silver 4210R @2.4GHz, 32 GB RAM and Nvidia Quadro RTX6000. The software infrastructure includes CUDA10.1, Python, PyTorch, LieTorch and Open3D.

### 4.1. Results on Matching

The quality of feature matching influences the performance of both tracking and mapping. To validate the superiority of network-based feature matching, matching results from SVR-Net are compared with results from the traditional method. The results are presented in both 3D and 2D visualizations, and the quality of feature matching is quantified using the end-point-error (EPE) metric.

The visualization of feature matching in 3D form is shown in Figure 4 and Figure 5. The feature matching is performed in a test scene of ScanNet. The blue-colored point clouds represent the central coordinates of voxels. Each voxel is implicitly attached with a feature vector obtained through voxel feature extraction. The black lines are match lines connecting voxels with their matched estimations of projection coordinates on the reference frame. Each pair of panels represents the same match from different perspectives. The left panel shows the pose of the reference frame and the RGB image represented by a point lattice. The right panel shows the result observed from the back of the reference frame, where the voxels in the environment almost overlap with their projection on the image.

As illustrated in Figure 4, the match lines of SVR-Net are consistent and conform to the projection rule, where voxels representing an object (such as the wardrobe) are projected onto the corresponding object in the image. The match lines in the right panel have short projection lengths, indicating low end-point-error. This is attributed to the integration capability of the matching network for the features of adjacent voxels.

Figure 5 illustrates match lines obtained through brute-force search, which contain inconsistent parts that indicate outliers. Some of the match lines in the right figure exhibit significant projection length with high end-point-error. The result indicates that estimation from SVR-Net conforms to projection rules, while the estimation of a feature’s L2 distance in brute-force search has no constraints from adjacent voxels, and this method is prone to producing outliers. A conventional ratio test can remove outliers, but reduces the available matches at the same time, as shown in the following Figure 6 and Figure 7. All frames in the Figures are included in ScanNet test scenes.

As illustrated in Figure 6, SVR-Net produces dense matches between 2D image frames. To ensure clarity, only the top 200 matches from SVR-Net, possessing the highest confidence weights, are represented. There are no outliers in the matches, and correct matches can be obtained even in low-texture regions. It can be observed that SVR-Net identified the lines in the scene through high-confidence matches. In the matching network, the position information of the lines is propagated to adjacent low-texture regions, enabling the network to match correctly in these areas and improving its robustness.

For comparison, Figure 7 illustrates the classical approach, which involves the extraction of SIFT feature points and their matches through brute-force search. To eliminate outliers, only matches with a test ratio greater than 0.9 from the classical approach are retained. However, some outliers still remain, due to repetitive features in the low-texture regions, which interfere with pose estimation and reduce robustness. Furthermore, the sparse matches preclude the generation of a dense map.

EPE is computed to evaluate the quality of feature matching. For a set of *k* feature points in a key frame, we denote their estimated matching coordinates at the corresponding reference frame as $x_{1},x_{2},\ldots ,x_{k}$, and their true matching coordinates as ${\hat{x}}_{1},{\hat{x}}_{2},\ldots ,{\hat{x}}_{k}$. Then, the end-point-error is
(6)$$\frac{1}{k}\sum_{i=1}^{k}{∥x_{i}-{\hat{x}}_{i}∥}_{2}.$$

The EPE of a varying number of feature points is plotted in Figure 8. The evaluation involves 304 pairs of frames in two test scenes from the ScanNet dataset. In order to provide a given number of matched features, only matches with the highest confidence weights or test ratios are selected. As the number of matches increases, traditional SIFT-based methods become less effective in distinguishing repeated features, resulting in an increase in EPE. In contrast, SVR-Net experiences a decrease in EPE as the measurement quantity increases. When there are numerous matches, the EPE of SVR-Net is significantly lower than that of traditional methods, indicating that it is more robust for dense matching.

### 4.2. Results on Full System

SVR-Net estimates both raw and fine partial maps for every pair of adjacent frames. The partial dense TSDF maps of a test scene in ScanNet are shown in Figure 9. The maps in (a) and (b) are estimated maps with voxel sizes of 16 cm and 8 cm, and the map in (c) is the ground-truth map with a voxel size of 4 cm. The RGB image in (d) is the corresponding key frame.

The full SLAM results on scene 360 of the TUM-RGBD dataset are shown in Figure 10. Camera trajectory and global maps are processed incrementally in the SLAM system. The estimated trajectory is displayed in blue, while the ground truth is in black. The dense global map is showcased by both the TSDF map and the point cloud of voxel coordinates at a voxel size of 8 cm. It can be observed that the pose estimated by the network almost coincides with the ground truth pose. The TSDF map in the left panel illustrates the floor and wall of the environment. The right panel presents the distribution of environment points through a dense point cloud.

The monocular SLAM system is evaluated on nine indoor scenes from the TUM-RGBD dataset. For each scene, the system runs on sequential RGB frames and outputs the result of tracking and dense mapping. The tracking performance is compared with those of ORB-SLAM and DeepV2D. All trajectories are evaluated up to scale using absolute trajectory error (ATE), which computes the rooted mean square translation error between estimated and ground truth poses. The result is shown in Table 2. All methods are provided mono. In challenging scenarios, the conventional ORB SLAMs fail, whereas deep approaches exhibit greater robustness, effectively computing poses in all scenes. Furthermore, the SVR-Net model outperforms DeepV2D with respect to average ATE. It is noteworthy that, in situations where ORB SLAMs achieve success, learning-based methods are less accurate due to the generalization error of networks. However, in the context of learning-based approaches, SVR-Net, which only estimates the relative pose between adjacent frames, achieves a similar ATE to that of DeepV2D, which utilizes multi-frame joint optimization. This suggests that SVR-Net demonstrates lower generalization error.

## 5. Discussion

Deep learning methods have been applied to the processes of monocular SLAM or dense TSDF mapping in various systems. However, there have been a limited number of studies integrating both of these aspects in a unified network. The suggested system, which is capable of simultaneously estimating pose and TSDF map using an end-to-end network, is more suitable for applications that require both aspects. The same features are exploited to produce two modalities of outputs with accuracy comparable to that of DeepV2D, which indicates highly efficient data utilization.

Real-time performance is an important metric for SLAM. SVR-Net runs at a frame rate of 4Hz on a desktop RTX2080. However, its performance on embedded devices or decentralized systems has not been tested, and this requires further research in the future.

SVR-Net only uses information from adjacent frames, which constrains the accuracy of localization and propagates errors to the map, thereby reducing its precision. A pose graph can significantly reduce pose error using multi-frame joint optimization. Additional research is necessary in the future to incorporate this into the proposed network.

## 6. Conclusions

This paper proposes SVR-Net, an end-to-end network for monocular TSDF SLAM. SVR-Net produces both poses and global TSDF maps from a sequence of monocular images. A metric encoder is constructed for the purpose of measuring image similarities, while a semantic encoder is devised to encode information, including the scale of scenes, to facilitate matching. In the matching network, sparse 3D convolution is adopted for dense TSDF mapping. A GRU is also employed to enable iterative updates for rectifying matches. Matching results presented in both 3D and 2D visualizations demonstrate that this approach effectively eliminates outliers, producing matches which are dense and more consistent than those produced by the traditional SIFT-based matching method. An experiment using the TUM-RGBD dataset further validates the conclusion that SVR-Net exhibits greater robustness compared with traditional ORB SLAMs. SVR-Net integrates Gauss–Newton updates for accurate pose estimation. Our experiments demonstrate that SVR-Net achieves a level of localization accuracy comparable to that of DeepV2D. Unlike previous monocular SLAM methods, the network directly yields a dense TSDF map during localization, which avoids inconsistency from depth map fusion and is more feasible for downstream tasks that are highly dependent on dense maps. Compared with existing TSDF mapping methods, SVR-Net does not require pre-calibrated poses and can meet real-time requirements.

## Figures and Tables

**Figure 1 sensors-23-03942-f001:**
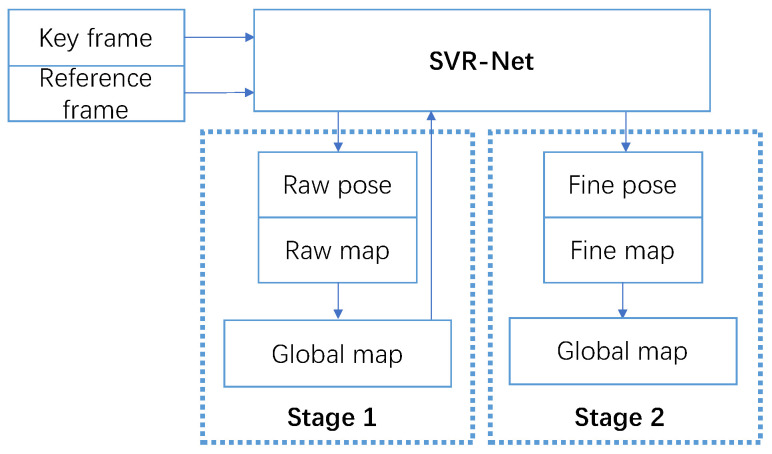
SLAM system.

**Figure 2 sensors-23-03942-f002:**
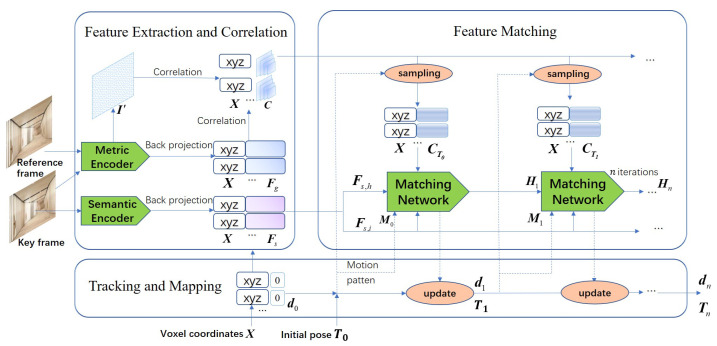
The structure of SVR-Net.

**Figure 3 sensors-23-03942-f003:**
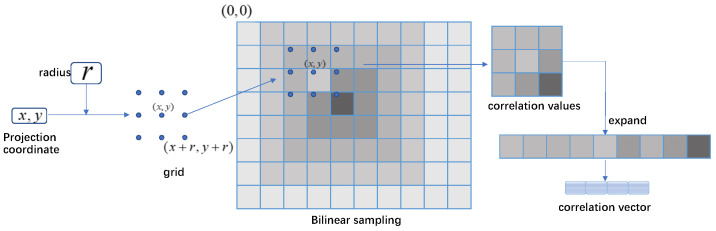
Sampling pipeline.

**Figure 4 sensors-23-03942-f004:**
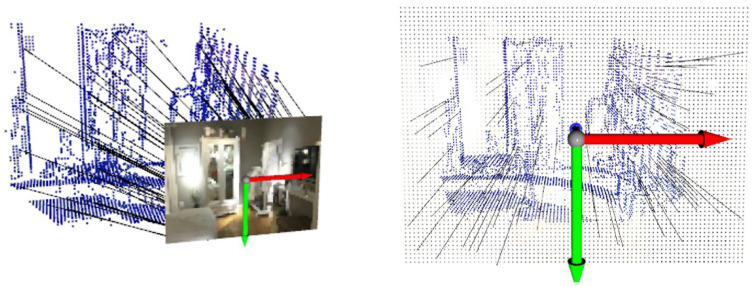
Matching result from SVR-Net(3D).

**Figure 5 sensors-23-03942-f005:**
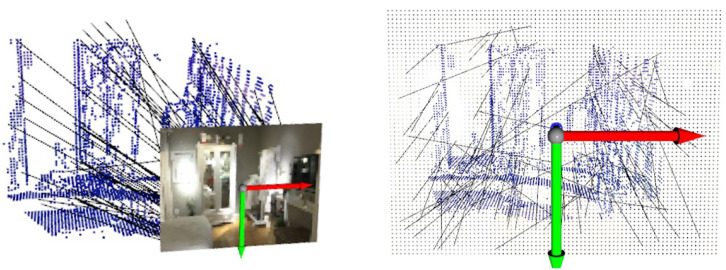
Matching result from brute-force search(3D).

**Figure 6 sensors-23-03942-f006:**
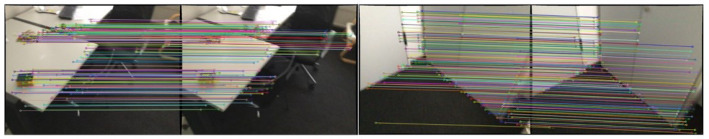
Matching result from SVR-Net(2D).

**Figure 7 sensors-23-03942-f007:**
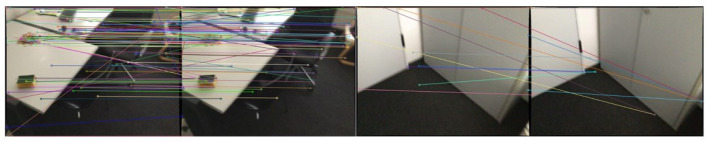
Matching result from SIFT(2D).

**Figure 8 sensors-23-03942-f008:**
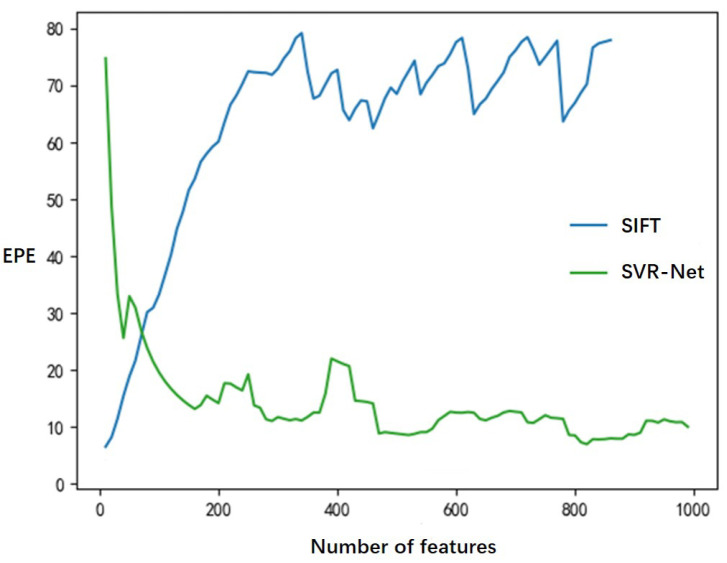
EPE at different numbers of feature points.

**Figure 9 sensors-23-03942-f009:**
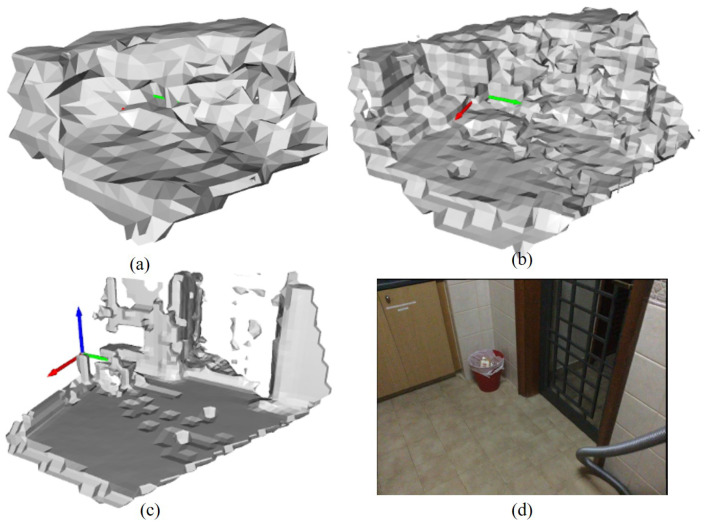
Partial dense TSDF maps and corresponding image. Estimated maps with voxel sizes of 16 cm and 8 cm are shown in (**a**,**b**) respectively. The ground-truth map with a voxel size of 4 cm is shown in (**c**). The RGB image from the corresponding key frame is in (**d**).

**Figure 10 sensors-23-03942-f010:**
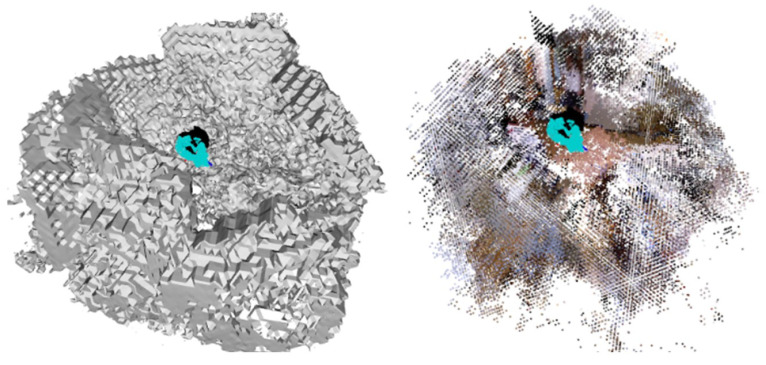
Full SLAM result.

**Table 1 sensors-23-03942-t001:** Comparison of existing learning-based approaches.

References	Year	Method	Monocular Input	Pose Estimation	TSDF Mapping
[28]	2020	DeepFactors	Yes	Yes	No
[35]	2020	D3VO	Yes	Yes	No
[14]	2020	DeepV2D	Yes	Yes	No
[32]	2021	IMAP	No	Yes	No
[15]	2021	DROID-SLAM	Yes	Yes	No
[18]	2021	NeuralRecon	Yes	No	Yes
[20]	2021	VoRTX	Yes	No	Yes
[46]	2021	VolumeFusion	Yes	No	Yes
[33]	2022	NICE-SLAM	No	Yes	Yes

**Table 2 sensors-23-03942-t002:** ATE on the TUM-RGBD benchmark. Failures are recorded as X.

Methods	360	Desk	Desk2	Floor	Plant	Room	rpy	Teddy	xyz	Average
ORB-SLAM2	X	0.071	X	0.023	X	X	X	X	0.01	-
ORB-SLAM3	X	0.017	0.21	X	0.034	X	X	X	0.009	-
DeepV2D	0.243	0.166	0.379	1.653	0.203	0.246	0.105	0.316	0.064	0.375
Ours	0.205	0.266	0.255	0.433	0.383	0.564	0.521	0.541	0.13	0.366

## Data Availability

Data sharing not applicable.

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
