# Peer review of "SVR-Net: A Sparse Voxelized Recurrent Network for Robust Monocular SLAM with Direct TSDF Mapping"

_sensors, 2023, doi:10.3390/s23083942_

Round 1

Reviewer 1 Report

The paper idea is valid and added knowledge to the literature. It is related to the journal scope. Overall, the paper is an accepted with major changes. The limitation of the paper lies in paper structures and its presentation as it has to be improved.

·     The related works aren’t enough where more related works have to be added. The reference is huge, but they aren't sufficient. Thus, a new recently published paper has to be cited. e.g.:

·       Zhao, L., Xu, S., Liu, L., Ming, D., & Tao, W. (2022). SVASeg: Sparse Voxel-Based Attention for 3D LiDAR Point Cloud Semantic Segmentation. Remote Sensing, 14(18), 4471.

·       Suryanarayana, G., Chandran, K., Khalaf, O. I., Alotaibi, Y., Alsufyani, A., & Alghamdi, S. A. (2021). Accurate Magnetic Resonance Image Super-Resolution Using Deep Networks and Gaussian Filtering in the Stationary Wavelet Domain. IEEE Access, 9, 71406-71417.

·       Liu, L., Gu, J., Zaw Lin, K., Chua, T. S., & Theobalt, C. (2020). Neural sparse voxel fields. Advances in Neural Information Processing Systems, 33, 15651-15663.

·     It will be a good idea of the authors can add a new table at the end of the second section related works to summarize and compare the existing approaches. The table should include the following columns: reference no, published year, approach name, advantages, and disadvantages. 

·     In results section, each table and figure have to be explained in separate paragraph.

·     The results fail to support the authors idea. So, you have to analysis the results in  more details.

·     The discussion is missing.

·     The conclusion section is very poor.

·     The paper limitations and future research direction are missing.

·     The paper has to be proofread. You have to reduce using the personal voice (we).

Author Response

We would like to express our special thanks to editor and the reviewers for taking the time to review the paper and for their valuable comments. We have carefully reviewed the referee report and have taken note of the areas for improvement. Modifications in the main text are marked in blue.

Point 1:     The related works aren’t enough where more related works have to be added. The reference is huge, but they aren't sufficient. Thus, a new recently published paper has to be cited. e.g.:

  • Zhao, L., Xu, S., Liu, L., Ming, D., & Tao, W. (2022). SVASeg: Sparse Voxel-Based Attention for 3D LiDAR Point Cloud Semantic Segmentation. Remote Sensing, 14(18), 4471.
  • Suryanarayana, G., Chandran, K., Khalaf, O. I., Alotaibi, Y., Alsufyani, A., & Alghamdi, S. A. (2021). Accurate Magnetic Resonance Image Super-Resolution Using Deep Networks and Gaussian Filtering in the Stationary Wavelet Domain. IEEE Access, 9, 71406-71417.
  • Liu, L., Gu, J., Zaw Lin, K., Chua, T. S., & Theobalt, C. (2020). Neural sparse voxel fields. Advances in Neural Information Processing Systems, 33, 15651-15663.

Response 1:     References to SVASeg and the recent SLAM work have been added to the related work section, line 113.

Point 2:     It will be a good idea of the authors can add a new table at the end of the second section related works to summarize and compare the existing approaches. The table should include the following columns: reference no, published year, approach name, advantages, and disadvantages.

Response 2:     A new table has been generated at the end of the related work section. Advantages and disadvantages of the existing approaches are summarized in a comparative way.

Point 3:     In results section, each table and figure have to be explained in separate paragraph.

Response 3:     The images in the results section have been rearranged so that each image is explained in a separate paragraph.

Point 4:     The results fail to support the authors idea. So, you have to analysis the results in more details.

Response 4:     More detailed explanations have been provided for most of the experimental results to better support the conclusions about the network's robustness.

Point 5:     The discussion is missing.

Response 5:     The discussion section has been added in which we analyze the limitations and future research directions.

Point 6:     The conclusion section is very poor.

Response 6:     The conclusion section has been enriched stating more finding of this research.

Point 7:     The paper limitations and future research direction are missing.

Response 7:     The paper limitations and future research directions have been added to the discussion section.

Point 8:     The paper has to be proofread. You have to reduce using the personal voice (we).

Response 8:     The paper has been proofread. The use of the personal voice has been reduced.

Reviewer 2 Report

1)      Authors should add motivation and contribution in the introduction section

2)      Author should make one table in related study with existing work and their limitations?

3)      How the proposed model is beneficial in decentralized system?

4)      All the table must be improved and the text within the table must be aligned properly.

5)      The grammar and typos error must be taken care 

6)      Author should add advantages and disadvantages of the proposed model.

7)      Is the proposed system secure enough and sutainable to apply in distributed environment. If yes, kindly approach with the below work and preferably include in the realted work

a.      Permissioned blockchain and deep-learning for secure and efficient data sharing in industrial healthcare systems. IEEE Transactions on Industrial Informatics.

b.       P2tif: a blockchain and deep learning framework for privacy-preserved threat intelligence in industrial iot. IEEE Transactions on Industrial Informatics.

Author Response

We would like to express our special thanks to editor and the reviewers for taking the time to review the paper and for their valuable comments. We have carefully reviewed the referee report and have taken note of the areas for improvement. Modifications in the main text are marked in blue.

Point 1:    Authors should add motivation and contribution in the introduction section.

Response 1:    Motivation has been enriched in line 23, and contribution has been added in line 77.

Point 2:    Author should make one table in related study with existing work and their limitations?

Response 2:    A new table has been generated at the end of the related work section. Advantages and disadvantages of the existing approaches are summarized in a comparative way.

Point 3:    How the proposed model is beneficial in decentralized system?

Response 3:    The TSDF map of the system can be managed in a distributed manner, but we have not implemented this scheme and verified its performance yet.

Point 4:    All the table must be improved and the text within the table must be aligned properly.

Response 4:    The tables have been adjusted and now use uniform center alignment.

Point 5:    The grammar and typos error must be taken care.

Response 5:    The paper has been proofread.

Point 6:    Author should add advantages and disadvantages of the proposed model.

Response 6:    A discussion section has been added in which we analyze the limitations and future research directions.

Point 7:    Is the proposed system secure enough and sutainable to apply in distributed environment. If yes, kindly approach with the below work and preferably include in the realted work

  1. Permissioned blockchain and deep-learning for secure and efficient data sharing in industrial healthcare systems. IEEE Transactions on Industrial Informatics.
  2. P2tif: a blockchain and deep learning framework for privacy-preserved threat intelligence in industrial iot. IEEE Transactions on Industrial Informatics.

Response 7:    Unfortunately, this work did not consider security. This is an underexplored direction in SLAM and deserves further research.

Reviewer 3 Report

Comments on the Article

SVR-Net: A Sparse Voxelized Recurrent Network for Robust Monocular SLAM with Direct TSDF Mapping

This paper introduces SVR-Net, an end-to-end network for monocular TSDF SLAM. SVR-Net is more robust compared with traditional ORB SLAMs. And its localization accuracy is comparable with other learning-based methods. Moreover, the network simultaneously produces dense TSDF map during localization, which avoids inconsistency from depth map fusion and is more feasible for downstream tasks highly dependent on dense maps.

The authors may be advised to do the following major revisions before its publication:

1.     Polish the abstract and write it in precise manner with two or three important findings included over there.

2.     Highlight the limitations of the research?

3.     Add more details to the description of problem and explain the terms.

4.     More physical justifications would be better to help the audience in better understanding.

5.     Literature survey can be improved. Add some most recent articles may please be referenced and arxiv preprints be removed.

6.     Remove the grammatical error and all typo mistakes.  Many sentences start from And is not a good English. Also page 2 line 60 “resent”. Check throughout.

7.     Conclusion section can also be elaborated by clearly stating the finding of this research and comparing them with the existing research 

Author Response

We would like to express our special thanks to editor and the reviewers for taking the time to review the paper and for their valuable comments. We have carefully reviewed the referee report and have taken note of the areas for improvement. Modifications in the main text are marked in blue.

Point 1:    Polish the abstract and write it in precise manner with two or three important findings included over there.

Response 1:    The abstract is polished and contribution is highlighted at the end of it.

Point 2:    Highlight the limitations of the research?

Response 2:    Discussion section has been added in which we analyze the limitations and future research directions, including performance on embedded devices and usage of pose graph.

Point 3:    Add more details to the description of problem and explain the terms.

Response 3:    A more detailed description of the problem has been provided in line 25.

Point 4:    More physical justifications would be better to help the audience in better understanding.

Response 4:    More detailed explanations have been provided for most of the experimental results, including line 298, 304, 317 and 325 in the results section.

Point 5:    Literature survey can be improved. Add some most recent articles may please be referenced and arxiv preprints be removed.

Response 5:    References to more recent work have been added to the related Work section in line 113. Arxiv preprints have been removed or replaced.

Point 6:    Remove the grammatical error and all typo mistakes.  Many sentences start from And is not a good English. Also page 2 line 60 “resent”. Check throughout.

Response 6:    We have have made revisions to address the grammatical errors and typos. We have also corrected the sentence structure to avoid starting sentences with "And". Additionally, we have corrected the spelling mistakes.

Point 7:    Conclusion section can also be elaborated by clearly stating the finding of this research and comparing them with the existing research.

Response 7:    The conclusion section has been enriched according to the suggestions. We appreciate your attention to detail and your valuable feedback.

Reviewer 4 Report

The authors propose in this paper a monocular SLAM system based on their sparse voxelized recurrent network, namely, SVR-Net; the model uses a pair of frames as input.

SVR-Net extracts voxel features for correlation and recursively matches them to estimate pose and dense map. The authors state that the sparse voxelized structure is designed to reduce memory usage of voxel features. They adopted gated recurrent unit to iteratively search for optimal match on correlation maps, which improves robustness. They embedd Gauss-Newton updates in iterations to impose geometrical constrain, which ensures accuracy of pose estimation.

SVR-Net is evaluated on TUM-RGBD benchmark and successfully estimates poses on all scenes.

They report that traditional ORB-SLAM fails on most of the same examples.

Figure 2 is quite important and also illustrative - however, the explanation about the figure should be more elaborated.

I would have liked to see a little explanation about the performance of the system in regard to execution time. May be the authors could add some content about this in the final manuscript.

Author Response

We would like to express our special thanks to editor and the reviewers for taking the time to review the paper and for their valuable comments. We have carefully reviewed the referee report and have taken note of the areas for improvement. Modifications in the main text are marked in blue.

Point 1:    Figure 2 is quite important and also illustrative - however, the explanation about the figure should be more elaborated.

Response 1:    The explanation of Figure 2 has been enriched to further explain each step in the figure.

Point 2:    I would have liked to see a little explanation about the performance of the system in regard to execution time. May be the authors could add some content about this in the final manuscript.

Response 2:    The discussion section has been added in which we report a frame rate of 4Hz on a desktop RTX2080.

Round 2

Reviewer 1 Report

The related works aren’t enough where more related works have to be added. The reference is huge, but they aren't sufficient. Thus, a new recently published paper has to be cited. e.g.:

  • Suryanarayana, G., Chandran, K., Khalaf, O. I., Alotaibi, Y., Alsufyani, A., & Alghamdi, S. A. (2021). Accurate Magnetic Resonance Image Super-Resolution Using Deep Networks and Gaussian Filtering in the Stationary Wavelet Domain. IEEE Access, 9, 71406-71417.
  • Liu, L., Gu, J., Zaw Lin, K., Chua, T. S., & Theobalt, C. (2020). Neural sparse voxel fields. Advances in Neural Information Processing Systems, 33, 15651-15663.

Author Response

We would like to express our special thanks to editor and the reviewers for taking the time to review the paper and for their valuable comments. We have carefully reviewed the referee report and have taken note of the areas for improvement. Modifications in the main text are marked in blue.

Point 1: The related works aren’t enough where more related works have to be added. The reference is huge, but they aren't sufficient. Thus, a new recently published paper has to be cited. e.g.:

  • Suryanarayana, G., Chandran, K., Khalaf, O. I., Alotaibi, Y., Alsufyani, A., & Alghamdi, S. A. (2021). Accurate Magnetic Resonance Image Super-Resolution Using Deep Networks and Gaussian Filtering in the Stationary Wavelet Domain. IEEE Access, 9, 71406-71417.
  • Liu, L., Gu, J., Zaw Lin, K., Chua, T. S., & Theobalt, C. (2020). Neural sparse voxel fields. Advances in Neural Information Processing Systems, 33, 15651-15663.

Response 1: The references to the recently published papers have been added in line 41 and 114.

Reviewer 2 Report

The authors have incorporated the changes successfully

Author Response

Thanks again for taking the time to review the paper and for their valuable comments. We have carefully reviewed the referee report from all reviewers and have taken note of the areas for improvement. Modifications in the main text are marked in blue on line 41 and 114. 

Reviewer 3 Report

Authors have revised the paper according to comments

Author Response

Thanks again to editor and the reviewers for taking the time to review the paper and for their valuable comments. We have carefully reviewed the referee report from all reviewers and have taken note of the areas for improvement. Modifications in the main text are marked in blue on line 41 and 114.